**Data Availability Statement:** All relevant data are within the manuscript and its Supporting Information files.

# Diabetic foot ulcers: Retrospective comparative analysis from Sicily between two eras

**Valentina Guarnotta**[1☉]*, **Stefano Radellini**[1☉], **Enrica Vigneri**[1☉], **Achille Cernigliaro**[2], **Felicia Pantò**[1], **Salvatore Scondotto**[2], **Piero Luigi Almasio**[3], **Giovanni Guercio**[4], **Carla Giordano**[1]*

**1** Sezione di Malattie Endocrine, del Ricambio e della Nutrizione, Dipartimento di Promozione della Salute, Materno-Infantile, Medicina Interna e Specialistica di Eccellenza "G. D'Alessandro" (PROMISE), Università di Palermo, Palermo, Italy, **2** Osservatorio Epidemiologico, Dipartimento Salute, Regione Sicilia, Italia, **3** Sezione di Gastroenterologia ed Epatologia, Dipartimento di Promozione della Salute, Materno-Infantile, Medicina Interna e Specialistica di Eccellenza "G. D'Alessandro", PROMISE, Università degli Studi di Palermo, Palermo, Italy, **4** Sezione di Chirurgia d'Urgenza, Dipartimento di Chirurgia, Oncologia e Scienza Orale, DICHIRONS, Università degli Studi di Palermo, Palermo, Italy

☉ These authors contributed equally to this work.
* carla.giordano@unipa.it (CG); valentina.guarnotta@unipa.it (VG)

## Abstract

### Aim

The aim of this study was to analyze changes in the incidence, management and mortality of DFU in Sicilian Type 2 diabetic patients hospitalized between two eras, i.e. 2008–2013 and 2014–2019.

### Methods

We compared the two eras, era1: 2008–13, era2: 2014–19. In era 1, n = 149, and in era 2, n = 181 patients were retrospectively enrolled.

### Results

In the population hospitalized for DFU in 2008–2013, 59.1% of males and 40.9% of females died, whilst in 2014–2019 65.9% of males and 34.1% of females died. Moderate chronic kidney disease (CKD) was significantly higher in patients that had died than in ones that were alive (33% vs. 43%, p < 0.001), just as CKD was severe (14.5% vs. 4%, p < 0.001). Considering all together the risk factors associated with mortality, at Cox regression multivariate analysis only moderate-severe CKD (OR 1.61, 95% CI 1.07–2.42, p 0.021), age of onset greater than 69 years (OR 2.01, 95% CI 1.37–2.95, p <0.001) and eGFR less than 92 ml/min (OR 2.84, 95% CI 1.51–5.34, p 0.001) were independently associated with risk of death.

### Conclusions

Patients with DFU have high mortality and reduced life expectancy. Age at onset of diabetic foot ulcer, eGFR values and CKD are the principal risk factors for mortality.

**Funding:** This research received specific funds from the Sicilian Health Department "PSN Piede Diabetico" that contributed to create the Diabetic Foot Centre in 2013; these funds were provided in the form of a grant (No. 37475) awarded to CG.

**Competing interests:** The authors have declared that no competing interests exist.

## Introduction

Diabetic foot ulcers (DFU) are one of the most common complications among patients with diabetes and are associated with significant morbidity and mortality [1–3]. The American Diabetes Association (ADA) estimates that 20–35% of diabetic patients develop DFU in their lifetime and for this reason prevention is crucial to DFU management [4]. Many underlying factors are recognized to account for reduction of DFU onset and progression [5–7]. However, strict glycaemic control is seen as the principal factor influencing possible healing of DFU [8, 9]. Atherosclerosis is the principal risk factor for ischemia whilst neuropathy with its specific symptoms, i.e. diminished or complete loss of protective sensation, paresthesia and burning, is the principal condition responsible for the development of foot ulcers [10].

The median time for healing is 12 weeks and 5-year survival following presentation with a new DFU is estimated to be around 50–60%. In this respect, 1-, 2-, and 5-year survival only proves to occur in 81%, 69% and 29% of cases, respectively [11, 12]. The risk of death at 5 years for a patient with DFU is 2.5 times as high as the risk for a patient with diabetes who does not have a foot ulcer [10]. Indeed, occurrence of DFU is an independent predictor of mortality even at 10 years. The link between DFU and renal failure is well recognized, just as there is a well-known temporal relationship between DFU and the onset of dialysis for End Stage Renal disease (ESDR) [10, 12, 13]. In addition, inflammation associated with ulceration can trigger the final decline in renal function [10, 14, 15].

Diabetic foot complications are serious and expensive. Furthermore, DFU is associated with prolonged hospitalization, especially when the lower extremities are amputated [16, 17].

The incidence of major amputation is used as a surrogate for failure of DFU to heal. Currently, regarding mortality, diabetic foot disease is considered analogous to malignancy and among the multiple factors predisposition to ulceration is dependent on neuropathy and PAD and the trigger factor is trauma [16–18].

Although DFU most often result from the combination of the two major complications of diabetes, diabetic neuropathy and arterial disease, it can be complicated by soft tissue and bone infection. Arterial disease represents the most severe prognostic factor in terms of amputation and survival. Prevention remains the most effective strategy against DFU and assessing improvement in the management of diabetes and its complications based on the evolution of hospitalization rates for DFU and lower extremity amputation in individuals with diabetes is fundamental for prevention [10, 16–19].

The aim of this study was to analyze changes in the incidence of DFU, and the evolution of hospitalization and management of DFU between two eras, i.e. 2008–2013 and 2014–2019. In this light, the primary objective of our study was to establish the mortality rate in the entire cohort of type 2 diabetic patients hospitalized for DFU in our department and after in the two groups divided for the two eras. As a secondary objective, we aimed to better understand whether the characteristics and outcomes of DFU patients changed in the two periods, seeing that in the second era a dedicated diabetic foot team was established in our clinic. Actually, the two periods correspond to a phase of improvement and standardization in the medical management of DFU according to the National Institute of Health and Care Excellence guidelines on management of DFU, which were changed in 2004, and updated in 2010 and 2016 [20].

## Materials and methods

We retrospectively reviewed the medical records of 330 consecutive hospitalized patients with DFU from 2008 to 2019 at the Division of Endocrinology and Diabetology of AOUP Paolo Giaccone, University of Palermo, Italy. At our centre, in 2013 a Diabetic Foot Centre was created utilizing funds from the Sicilian Health Department, which permitted the establishment

of a multidisciplinary foot care team (MDFT) where diabetologists had the principal responsibility, and general surgeons, vascular surgeons, infectious specialists, cardiologists, orthopedists, radiologists, microbiologists, podiatrists and diabetes nurse educators were also included. For this reason, the data were collected considering the patients hospitalized from 2008 to 2013 and from 2014 to the end of 2019, separately. The baseline characteristics of the diabetic patients were grouped and mortality data were kindly provided by the Epidemiological Observatory of the Sicily Region through the patients' tax codes and year of hospitalization.

Ischemic heart disease and heart failure were considered as cardiovascular diseases. The presence of Chronic Kidney Disease (CKD), dyslipidemia, arterial hypertension, systemic inflammatory response syndrome (SIRS), peripheral vascular disease and retinopathy classes were defined according to the most recent international guidelines [20–25]. The University of Texas systems (UT classification) were used to classify the severity of ulcers [26]. Successful revascularization was defined in patients who underwent percutaneous transluminal angioplasty (PTA) [27].

The clinical data and ulcer-related outcomes in the two cohorts, comprising a total of 330 patients, were compared with data obtained during the periods 2008–2013 (N = 149) and 2014–2019 (N = 181). Diabetic therapy was distinguished by the use of oral hypoglycemic agents, basal-bolus insulin or the combination of oral hypoglycemic therapy plus basal insulin. Antiplatelet and hypolipidemic therapies were also considered. Amputations were divided into minor and major amputations. Complete wound healing was defined as the complete epithelialization of the overlying soft tissue wound after admission. Exclusion criteria were the following: more than two DFU recurrences in the last 3 years, previous > 5 years DFU in the other foot, cachexia and age over 90 years.

Every patient received appropriate multi-disciplinary care including bed rest, wound debridement, daily wound dressing, antibiotic therapy, skin grafting and limited amputation, control of blood glucose and treatment of associated comorbidities. Follow-up was continued until the patients were discharged from hospital and came as outpatients, or else died.

The study was approved by the Local Ethical Committee and carried out in accordance with the Declaration of Helsinki for experiments involving humans. At the time of observation all patients, regularly informed of the aim of the study, signed an informed consent for scientific use of their data.

## Statistical analysis

SPSS version 17 and MedCalc version 11.3 were used for data analysis. Baseline characteristics were presented as mean ± SD for continuous variables; rates and proportions were calculated for categorical data. Normality of distribution for quantitative data was assessed by the Shapiro-Wilk test. The differences between dead and alive and between hospitalized in the periods 2008–2013 and 2014–2019 were detected by Student's t test for continuous variables and by the chi-square test for categorical variables. Kaplan-Meier survival curves were compared using log-rank test. Crude odds ratios (OR) and their 95% CI for the association of mortality with potential risk factors in patients with DFU were calculated by univariate analysis. Predictors that were associated with the outcomes with a p-value <0.05 were entered in a multivariate analysis. Cox proportional hazards regression was used to estimate hazard ratios for all-cause deaths. The receiver operating characteristic (ROC) analysis was performed to investigate the diagnostic ability of significantly associated risk factors to predict mortality. The ROC curve is plotted as sensitivity versus 1-specificity. The area under the ROC curve (AUC) was estimated to measure the overall performance of the predictive factors of mortality. A p value of <0.05 was considered statistically significant.

## Results

Two hundred and nineteen males and 111 females were hospitalized in the study periods 2008–2013 (45.1%) and 2014–2019 (54.8%), respectively. Sixty percent of the Type 2 diabetic patients hospitalized in 2008–2013 and 40% of those hospitalized in 2014–2019 died (p<0.001) mainly due to cardiovascular disease (coronary artery disease; myocardial infarction; cardiac arrest or other cardiac causes), bronchopneumonia, cancer, cerebrovascular accidents, renal failure, pulmonary thromboembolic disease, gastrointestinal bleeding and other causes.

The clinical characteristics after hospitalization of DFU patients who later dies or are still alive are shown in Table 1. Arterial hypertension was more frequent in patients who died (94.5%) than in ones still living (80.9%, p<0.001). Myocardial infarction was more frequent in diabetic patients who died (42.7%) in comparison to 26.3% in living ones (p = 0.003). Current smoking was more frequent in patients who died than in ones still alive (p = 0.030). Moderate chronic kidney disease (CKD) was significantly higher in patients who died than living ones (33% vs. 43%, p<0.001), just as CKD was more severe (14.5% vs. 4%, p<0.001). Peripheral vascular disease was more frequent in patients who died than in ones still alive (62.7 vs. 18.6%, p = 0.035). Neuropathic lesions were less frequent in patients who died than in ones still alive (22.8% vs. 37.3%, p = 0.005).

In the cohort of hospitalized dead patients stage D Texas ulcers were more frequent than in ones still alive (60.9 vs. 46.4%, p = 0.009), while stage B was less frequent in dead than living (34.5 vs. 48.2%, p = 0.012). The mean age of patients with diabetes hospitalized for foot ulcers was slightly higher in patients who dies than in living patients (p<0.001) (Table 2). Type 2 diabetic patients who dies showed higher duration of the disease (p = 0.012), higher creatinine values (p<0.001) and lower eGFR (p<0.001), in comparison to those who are still alive (Table 2). No differences were found as regards BMI, total healing time, lipids and inflammatory parameters confirming the same gravity of sepsis, requiring hospitalization (Table 2).

In the population hospitalized for DFU sepsis in 2008–2013 and in 2014–2019 periods, arterial hypertension (p = 0.011), dyslipidemia (p<0.001), mild chronic kidney disease (p = 0.013), mild, moderate non-proliferative and proliferative retinopathy (all p<0.001), oral hypoglycaemic agents (p = 0.013), combined oral hypoglycaemic agents and long-acting insulin (p = 0.016), revascularization treatment (p = 0.026) were more frequent in the 2008–2013 than 2014–2019 periods (Table 3). On the other side, stroke (p = 0.004), current (p<0.001) and former smoking (p<0.001), peripheral vascular disease (p = 0.001), basal bolus insulin (p<0.001), ischemic lesions (p = 0.013), osteomyelitis (p = 0.015), minor amputations (p = 0.019) were less frequent in the period 2008–2013 than 2014–2019 (Table 3). The comparison between patients who died in 2008–2013 and 2014–2019 showed that patients who died in the first era had higher frequency of dyslipidemia (p = 0.003), mild kidney disease (p = 0.019), mild, moderate and severe retinopathy (all p<0.001), hypolipidemic treatment (p = 0.003) and treatment with oral hypoglycaemic agents (p = 0.008) and combined oral hypoglycemic agents and insulin (p = 0.023) and lower frequency of stroke (p = 0.005), cardiac insufficiency (p = 0.025), former smoker (p<0.001), treatment with basal-bolus insulin (p = 0.001), ischemic lesion type (p = 0.017), dorsal lesion (p = 0.005) and grade 3 (p = 0.021) than second era (Table 4). In addition, patients who died in the first era had higher serum total cholesterol values (p = 0.001) and lower serum creatinine (p = 0.021) than patients who died in the second era (Table 5).

In the population hospitalized for DFU in 2008–2013 (n 66 out of 149), 59.1% of males and 40.9% of females died, whilst in 2014–2019 (n = 181) 65.9% of males and 34.1% of females died. Among patients hospitalized in the period 2008–2013, dead patients have higher frequency of moderate (p = 0.016) and severe chronic kidney disease (p = 0.001), peripheral

**Table 1. General characteristics of all patients with diabetic foot complication.**

| | All patients (n = 330) | Dead (n = 110) | Alive (n = 220) | |
|---|---|---|---|---|
| | *Subjects (%)* | *Subjects (%)* | *Subjects (%)* | *p* |
| **Gender** | | | | |
| Males | 219 (22.7%) | 68 (61.8%) | 151 (68.7%) | 0.220 |
| Females | 111 (63.8%) | 42 (38.2%) | 69 (31.3%) | |
| **Hospitalization period** | | | | |
| 2008–2013 | 149 (45.1%) | 66 (60%) | 83 (37.7%) | <0.001 |
| 2014–2019 | 181 (54.8%) | 44 (40%) | 137 (62.2%) | <0.001 |
| **Arterial hypertension** | 282 (85.4%) | 104 (94.5%) | 178 (80.9%) | 0.001 |
| **Dyslipidemia** | 247 (74.8%) | 88 (80%) | 159 (72.3%) | 0.127 |
| **Cardiovascular disease** | | | | |
| Myocardial infarction | 105 (31.8%) | 47 (42.7%) | 58 (26.3%) | 0.003 |
| Stroke | 20 (6%) | 7 (6.3%) | 13 (5.9%) | 0.870 |
| Cardiac insufficiency | 1 (0.3%) | 1 (0.9%) | 0 | 0.758 |
| **Smoking** | | | | |
| Current | 53 (25.1%) | 29 (26.3%) | 24 (10.9%) | 0.030 |
| Former | 58 (17.5%) | 15 (13.6%) | 43 (19.5%) | 0.184 |
| **Chronic kidney disease** | | | | |
| Mild | 94 (28.4%) | 33 (30%) | 61 (27.7%) | 0.785 |
| Moderate | 76 (23%) | 33 (30%) | 43 (19.5%) | <0.001 |
| Severe | 25 (7.5%) | 16 (14.5%) | 9 (4%) | <0.001 |
| **SIRS** | 54 (16.3%) | 22 (20%) | 32 (14.5%) | 0.207 |
| **Peripheral vascular disease** | 110 (33.3%) | 69 (62.7%) | 41 (18.6%) | 0.035 |
| **Retinopathy** | | | | |
| Mild non-proliferative | 48 (14.5%) | 15 (13.6%) | 33 (15%) | 0.740 |
| Moderate non-proliferative | 20 (6%) | 10 (9%) | 10 (9%) | 0.103 |
| Proliferative | 36 (10.9%) | 10 (9%) | 26 (11.8%) | 0.527 |
| **Hypolipidemic therapy** | 257 (77.8%) | 91 (82.7%) | 166 (75.4%) | 0.133 |
| **Antiplatelet therapy** | 289 (87.5%) | 101 (91.8%) | 188 (85.4%) | 0.099 |
| **Diabetic treatment** | | | | |
| Oral hypoglycaemic agents | 44 (13.3%) | 12 (10.9%) | 32 (14.5%) | 0.360 |
| Basal-bolus insulin | 214 (64.8%) | 78 (70.9%) | 136 (61.8%) | 0.103 |
| Oral hypoglycaemic agents + long-acting insulin | 72 (21.8%) | 20 (18.1%) | 52 (23.6%) | 0.258 |
| **Lesion type** | | | | |
| Ischaemic | 29 (8.8%) | 12 (10.9%) | 17 (7.7%) | 0.336 |
| Neuropathic | 107 (32.4%) | 25 (22.8%) | 82 (37.3%) | 0.005 |
| Neuroischaemic | 194 (58.8%) | 73 (66.3%) | 121 (55%) | 0.048 |
| **Affected foot** | | | | |
| Right | 158 (47.9%) | 54 (49.1%) | 104 (47.3%) | 0.755 |
| Left | 138 (41.8%) | 40 (36.3%) | 98 (44.5%) | 0.155 |
| Both | 34 (10.3%) | 16 (14.5%) | 18 (8.2%) | 0.052 |
| **Lesion area** | | | | |
| I toe | 41 (12.4%) | 14 (12.7%) | 27 (12.3%) | 0.906 |
| Distal extremities | 43 (13%) | 17 (15.5%) | 26 (11.8%) | 0.355 |
| Lateral plantar | 87 (26.3%) | 28 (25.5%) | 59 (26.8%) | 0.585 |
| Medial plantar | 130 (36%) | 41 (37.2%) | 89 (40.5%) | 0.685 |
| Calcanear | 22 (6.6%) | 6 (5.5%) | 16 (7.3%) | 0.533 |
| Dorsal | 7 (1.8%) | 4 (3.6%) | 3 (1.3%) | 0.860 |

*(Continued)*

**Table 1.** (Continued)

| | All patients (n = 330) | Dead (n = 110) | Alive (n = 220) | |
|---|---|---|---|---|
| | *Subjects (%)* | *Subjects (%)* | *Subjects (%)* | *p* |
| **Osteomyelitis** | 23 (6%) | 5 (4%) | 18 (8%) | 0.221 |
| **Revascularization treatment** | 88 (26.6%) | 35 (31.8%) | 53 (24%) | 0.147 |
| **Surgery treatment** | | | | |
| Minor amputation | 90 (27.2%) | 30 (27.2%) | 60 (27.2%) | 0.463 |
| Major amputation | 12 (3.6%) | 6 (5.4%) | 6 (2.7%) | 0.149 |
| **VAC therapy** | 99 (30%) | 36 (32.7%) | 63 (28.6%) | 0.448 |
| **Stage** | | | | |
| A | 9 (2.7%) | 2 (1.8%) | 7 (3.2%) | 0.375 |
| B | 144 (43.6%) | 38 (34.5%) | 106 (48.2%) | 0.012 |
| C | 7 (2.1%) | 3 (2.7%) | 4 (1.8%) | 0.429 |
| D | 169 (50.1%) | 67 (60.9%) | 102 (46.4%) | 0.009 |
| **Grade** | | | | |
| 0 | 1 (0.3%) | 0 | 1 (0.5%) | 0.667 |
| 1 | 127 (38.5%) | 35 (31.8%) | 92 (41.8%) | 0.050 |
| 2 | 167 (50.6%) | 61 (55.5%) | 106 (48.2%) | 0.129 |
| 3 | 34 (10.3%) | 14 (12.7%) | 20 (9.1%) | 0.201 |

vascular disease (p = 0.005), basal-bolus insulin therapy (p = 0.022) and lower frequency of oral hypoglycaemic therapy (p = 0.048) and neuropathic lesion (p = 0.031) than living patients (S1 Table).

In the period 2014–2019 patients that died had arterial hypertension (p = 0.013), myocardial infarction (p = 0.014) and antiplatelet therapy (p = 0.029) and lower frequency of neuropathic lesion (p = 0.007) than living ones (S1 Table).

**Table 2. Clinical, metabolic and inflammatory parameters in all patients with diabetic foot complication.**

| | All patients (n = 330) | Dead (n = 110) | Alive (n = 220) | |
|---|---|---|---|---|
| | *Mean ± SD* | *Mean ± SD* | *Mean ± SD* | *P* |
| *General parameters* | | | | |
| Age at onset of diabetic foot (years) | 65.3 ± 12.1 | 70.3 ± 10.7 | 62.8 ± 12.1 | <0.001 |
| BMI (kg/m$^2$) | 29.2 ± 4.62 | 24.5 ± 3.1 | 24.5 ± 3.1 | 0.737 |
| Duration of diabetes (years) | 19.3 ± 11.9 | 21.6 ± 12.6 | 18.1 ± 16.6 | 0.012 |
| Healing time (days) | 29.1 ± 19.6 | 28.6 ± 19.9 | 29.3 ± 19.5 | 0.744 |
| *Metabolic parameters* | | | | |
| Creatinine (mg/dL) | 1.22 ± 0.88 | 1.52 ± 1.10 | 1.07 ± 0.71 | <0.001 |
| eGFR (mL/min) | 73.6 ± 30.9 | 59.1 ± 28.9 | 80.8 ± 29.3 | <0.001 |
| Urinary albumin (g/24h) | 0.33 ± 0.58 | 0.41 ± 0.62 | 0.31 ± 0.56 | 0.186 |
| HbA1c (%) | 10 ± 0.95 | 9.83 ± 1.8 | 10.1 ± 2.02 | 0.278 |
| Total cholesterol (mmol/L) | 3.8 ± 1.04 | 3.74 ± 1.03 | 3.83 ± 1.04 | 0.477 |
| HDL cholesterol (mmol/L) | 0.89 ± 0.29 | 0.87 ± 0.31 | 0.89 ± 0.29 | 0.581 |
| LDL cholesterol (mmol/L) | 2.18 ± 0.88 | 2.11 ± 0.90 | 2.21 ± 0.87 | 0.316 |
| Triglycerides (mmol/L) | 1.58 ± 0.68 | 1.63 ± 0.69 | 1.55 ± 0.68 | 0.322 |
| *Inflammatory parameters* | | | | |
| VES (mm) | 47.1 ± 24.4 | 48.7 ± 25.7 | 46.2 ± 23.8 | 0.391 |
| PCR (mg/L) | 57.3 ± 55.5 | 57.5 ± 57.3 | 57.4 ± 51.5 | 0.989 |

**Table 3. General characteristics of all patients with diabetic foot complication divided according to the time of hospitalization.**

| | Patients hospitalized 2008–2013 (n = 149) | Patients hospitalized 2014–2019 (n = 181) | p |
|---|---|---|---|
| **Gender** | | | |
| Males | 97 (65.1%) | 122 (67.4%) | |
| Females | 52 (34.9%) | 59 (32.6%) | 0.373 |
| **Arterial hypertension** | 135 (90.6%) | 147 (81.2%) | 0.011 |
| **Dyslipidemia** | 127 (85.2%) | 120 (66.3%) | <0.001 |
| **Cardiovascular disease** | | | |
| Heart attack | 53 (35.6%) | 52 (28.7%) | 0.113 |
| Stroke | 3 (2%) | 17 (9.4%) | 0.004 |
| Cardiac insufficiency | 0 | 1 (1.6%) | 0.165 |
| **Smoking** | | | |
| Current | 34 (22.8%) | 50 (27.6%) | <0.001 |
| Former | 4 (2.7%) | 53 (29.3%) | <0.001 |
| **Chronic kidney disease** | | | |
| Mild | 52 (34.9%) | 42 (23.2%) | 0.013 |
| Moderate | 34 (22.8%) | 42 (23.2%) | 0.520 |
| Severe | 11 (7.4%) | 14 (7.7%) | 0.538 |
| **SIRS** | 24 (16.1%) | 30 (16.6%) | 0.515 |
| **Peripheral vascular disease** | 67 (45%) | 113 (62.4%) | 0.001 |
| **Retinopathy** | | | |
| Mild non-proliferative | 36 (24.2%) | 12 (6.6%) | <0.001 |
| Moderate non-proliferative | 17 (11.4%) | 3 (1.7%) | <0.001 |
| Proliferative | 0 | 35 (19.3%) | <0.001 |
| **Hypolipidemic therapy** | 127 (85.2%) | 130 (71.8%) | 0.002 |
| **Antiplatelet therapy** | 127 (85.2%) | 162 (89.5%) | 0.158 |
| **Diabetic treatment** | | | |
| Oral hypoglycaemic agents | 28 (18.8%) | 16 (8.8%) | 0.007 |
| Basal-bolus insulin | 80 (53.7%) | 134 (74%) | <0.001 |
| Oral hypoglycaemic agents + long-acting insulin | 41 (27.5%) | 31 (17.1%) | 0.016 |
| **Lesion type** | | | |
| Ischaemic | 7 (4.7%) | 22 (12.2%) | 0.013 |
| Neuropathic | 54 (36.2%) | 52 (28.7%) | 0.091 |
| Neuroischaemic | 87 (58.4%) | 107 (59.1%) | 0.491 |
| **Affected foot** | | | |
| Right | 67 (45%) | 91 (50.3%) | 0.198 |
| Left | 67 (45%) | 71 (39.2%) | 0.174 |
| Both | 15 (10.1%) | 18 (9.9%) | 0.557 |
| **Lesion area** | | | |
| I toe | 21 (14.1%) | 20 (11%) | 0.252 |
| Distal extremities | 25 (16.8%) | 18 (9.9%) | 0.049 |
| Lateral plantar | 72 (48.3%) | 101 (55.8%) | 0.107 |
| Medial plantar | 59 (39.6%) | 60 (33.1%) | 0.136 |
| Calcanear | 6 (4%) | 16 (8.8%) | 0.062 |
| Dorsal | 2 (1.3%) | 15 (8.3%) | 0.003 |
| **Osteomyelitis** | 5 (3.4%) | 18 (9.9%) | 0.015 |
| **Revascularization treatment** | 48 (32.2%) | 40 (22.1%) | 0.026 |

(*Continued*)

**Table 3.** (Continued)

| | Patients hospitalized 2008–2013 (n = 149) | Patients hospitalized 2014–2019 (n = 181) | p |
|---|---|---|---|
| **Surgery treatment** | | | |
| Minor amputation | 32 (20.6%) | 58 (31.2%) | 0.019 |
| Major amputation | 4 (2.6%) | 8 (4.3%) | 0.290 |
| **VAC therapy** | 40 (26.8%) | 59 (32.6%) | 0.155 |
| **Stage** | | | |
| A | 6 (4%) | 4 (2.2%) | 0.165 |
| B | 58 (38.9%) | 86 (47.5%) | 0.073 |
| C | 5 (3.4%) | 2 (1.1%) | 0.152 |
| D | 80 (53.7%) | 89 (49.2%) | 0.240 |
| **Grade** | | | |
| 0 | 1 (0.7%) | 1 (0.6%) | 0.452 |
| 1 | 55 (36.9%) | 72 (39.8%) | 0.338 |
| 2 | 84 (56.4%) | 93 (51.4%) | 0.365 |
| 3 | 9 (6%) | 15 (8.3%) | 0.423 |

In patients hospitalized in the period 2008–2013 higher total cholesterol (p<0.001) and LDL-cholesterol (p<0.001) and lower healing time (p<0.001), creatinine (p = 0.003), HDL-cholesterol (p<0.001), urinary albumin (p = 0.018), VES (p = 0.001) were found in all patients compared to 2014–2019 period (S2 Table).

Older age (p<0.001), higher duration of diabetes mellitus (p = 0.002) and creatinine values (p<0.001) and lower total-cholesterol (p = 0.033), HDL-cholesterol (p = 0.046) and LDL-cholesterol (p = 0.037), eGFR (p<0.001) were observed in patients who died than in those still living in the 2008–2013 period (S2 Table). In 2014–2019 older age (p<0.001), higher creatinine values (p<0.001) and lower eGFR (p<0.001) were found in patients who dies compared to ones still living (S2 Table).

Fig 1A displays the Kaplan-Meier curves for survival. Overall survival probabilities and survival probabilities were assessed. Survival probability for all patients after 12 years of follow-up was 53% (Fig 1B). In the period 2008–2013 the survival probability was lower than in the period 2014–2019 (Fig 1C).

Considering all together the risk factors associated with mortality, at Cox regression multivariate analysis only moderate-severe CKD (OR 1.61, 95% CI 1.07–2.42, p = 0.021), age of onset greater than 69 years (OR 2.01, 95% CI 1.37–2.95, p <0.001) and eGFR less than 92 ml/min (OR 2.84, 95% CI 1.51–5.342.84 (range 1.51–5.34 p = 0.001) were independently associated with risk of death (Fig 1D).

## Discussion

The present study followed a cohort of patients with diabetes mellitus and DFU for a period of 12 years. Our study is the first Sicilian study conducted in a socially and ethnically homogeneous population to examine the mortality outcomes in patients with DFU. In patients with diabetes, DFU is recognized to be a marker for high mortality [28, 29]. This is confirmed by multiple studies from all over the world reporting that half of all patients who develop DFU die within 5 years [1, 4, 10].

In our study we confirmed that patients with later stages of CKD and advanced diabetic nephropathy have a greater risk of complications and mortality. The degree of renal

**Table 4. General characteristics of dead patients with diabetic foot complication divided in the two periods of hospitalization.**

| | Dead 2008–2013 (n = 66) | Dead 2014–2019 (n = 44) | p |
|---|---|---|---|
| **Gender** | | | |
| Males | 39 (59.1%) | 29 (65.9%) | 0.448 |
| Females | 27 (40.9%) | 15 (34.1%) | |
| **Arterial hypertension** | 63 (95.5%) | 41 (93.2%) | 0.113 |
| **Dyslipidemia** | 57 (86.4%) | 31 (70.5%) | 0.003 |
| **Cardiovascular disease** | | | |
| Myocardial infarction | 28 (42.4%) | 19 (43.2%) | 0.184 |
| Stroke | 3 (4.5%) | 4 (9.1%) | 0.005 |
| Cardiac insufficiency | 0 | 1 (2.3%) | 0.025 |
| **Smoking** | | | |
| Current | 15 (22.7%) | 10 (22.7%) | 0.497 |
| Former | 2 (3%) | 12 (27.3%) | <0.001 |
| **Chronic kidney disease** | | | |
| Mild | 23 (34.8%) | 10 (22.7%) | 0.019 |
| Moderate | 21 (31.8%) | 12 (27.3%) | 0.934 |
| Severe | 10 (15.2%) | 6 (13.6%) | 0.904 |
| **SIRS** | 14 (21.2%) | 8 (18.2%) | 0.369 |
| **Peripheral vascular disease** | 38 (57.6%) | 31 (70.5%) | 0.002 |
| **Retinopathy** | | | |
| Mild non-proliferative | 14 (21.2%) | 1 (2.3%) | <0.001 |
| Moderate non-proliferative | 10 (15.2%) | 0 | <0.001 |
| Proliferative | 10 (9%) | 0 | <0.001 |
| **Hypolipidemic therapy** | 58 (87.9%) | 33 (75%) | 0.003 |
| **Antiplatelet therapy** | 58 (87.9%) | 43 (97.7%) | 0.050 |
| **Diabetic treatment** | | | |
| Oral hypoglycaemic agents | 8 (12.1%) | 4 (9.1%) | 0.008 |
| Basal-bolus insulin | 42 (63.6%) | 36 (81.8%) | <0.001 |
| Oral hypoglycaemic agents + long-acting insulin | 16 (24.2%) | 4 (9.1%) | 0.023 |
| **Lesion type** | | | |
| Ischaemic | 4 (6.1%) | 8 (18.2%) | 0.017 |
| Neuropathic | 18 (27.3%) | 6 (13.6%) | 0.146 |
| Neuroischaemic | 43 (65.2%) | 30 (68.2%) | 0.984 |
| **Affected foot** | | | |
| Right | 33 (50%) | 21 (47.7%) | 0.337 |
| Left | 24 (36.4%) | 16 (36.4%) | 0.293 |
| Both | 9 (13.6%) | 7 (15.9%) | 0.971 |
| **Lesion area** | | | |
| I toe | 9 (13.6%) | 5 (11.4%) | 0.404 |
| Distal extremities | 14 (21.2%) | 3 (6.8%) | 0.066 |
| Lateral plantar | 32 (48.5%) | 28 (63.6%) | 0.176 |
| Medial plantar | 26 (39.4%) | 12 (27.3%) | 0.225 |
| Calcanear | 1 (1.5%) | 5 (11.4%) | 0.081 |
| Dorsal | 2 (3%) | 4 (9.1%) | 0.005 |
| **Osteomyelitis** | 3 (4.5%) | 2 (4.5%) | 0.495 |
| **Revascularization treatment** | 26 (39.4%) | 9 (20.5%) | 0.016 |

(*Continued*)

**Table 4.** (Continued)

| | | Dead<br>2008–2013<br>(n = 66) | Dead<br>2014–2019<br>(n = 44) | p |
|---|---|---|---|---|
| **Surgery treatment** | | | | |
| | Minor amputation | 15 (22.7%) | 15 (34.1%) | 0.203 |
| | Major amputation | 3 (4.5%) | 3 (6.8%) | 0.495 |
| **VAC therapy** | | 21 (31.8%) | 15 (34.1%) | 0.257 |
| **Stage** | | | | |
| | A | 0 | 2 (4.5%) | 0.188 |
| | B | 21 (31.8%) | 17 (38.6%) | 0.117 |
| | C | 2 (3%) | 1 (2.3%) | 0.158 |
| | D | 40 (60.6%) | 24 (54.5%) | 0.414 |
| **Grade** | | | | |
| | 0 | 0 | 0 | 0.270 |
| | 1 | 25 (37.8%) | 17 (38.6%) | 0.594 |
| | 2 | 41 (62.1%) | 25 (56.8%) | 0.057 |
| | 3 | 4 (6.1%) | 7 (15.9%) | 0.021 |

impairment correlates strongly with the incidence and prevalence of DFU with an adjusted OR equal to 1.61. Wolf et al. reported that impaired renal function was an independent predictor of all-cause mortality and cardiovascular deaths [30]. In the current study, eGFR < 92 ml/min was found to be a predictor of mortality with an OR of 2.84. These results are in line with those obtained by Ghanassia et al. who demonstrated that CKD was the only independent predictor of mortality in patients with DFU [31]. Similarly, in our cohort of Sicilian Type 2

**Table 5. Clinical, metabolic and inflammatory parameters in dead patients with diabetic foot complication divided according period of hospitalization.**

| | Dead<br>2008–2013<br>(n = 66)<br>*Mean ± SD* | Dead<br>2014–2019<br>(n = 44)<br>*Mean ± SD* | p |
|---|---|---|---|
| *General Parameters* | | | |
| Age at onset of diabetic foot (years) | 68.9 ± 10.3 | 72.5 ± 10.9 | 0.085 |
| BMI (kg/m$^2$) | 29.3 ± 11.7 | 28.6 ± 4.91 | 0.415 |
| Duration of diabetes (years) | 20.5 ± 12.2 | 23.3 ± 13.8 | 0.238 |
| Healing time (days) | 26.6 ± 13.5 | 31.4 ± 26.8 | 0.217 |
| *Metabolic parameters* | | | |
| Creatinine (mg/dL) | 1.24 ± 0.59 | 1.94 ± 1.51 | 0.001 |
| eGFR (mL/min) | 63.5 ± 25.9 | 52.6 ± 32.1 | 0.063 |
| Urinary albumin (g/24h) | 0.31 ± 0.55 | 0.52 ± 0.69 | 0.092 |
| HbA1c (%) | 9.88 ± 1.71 | 9.76 ± 1.34 | 0.735 |
| Total cholesterol (mmol/L) | 3.93 ± 1.02 | 3.46 ± 11.7 | 0.021 |
| HDL cholesterol (mmol/L) | 0.92 ± 0.28 | 0.81 ± 0.31 | 0.060 |
| LDL cholesterol (mmol/L) | 2.22 ± 0.89 | 1.94 ± 0.89 | 0.115 |
| Triglycerides (mmol/L) | 1.70 ± 0.69 | 1.54 ± 0.71 | 0.243 |
| *Inflammatory parameters* | | | |
| VES (mm) | 46.2 ± 25.9 | 52.5 ± 25.1 | 0.207 |
| PCR (mg/L) | 60.2 ± 40.8 | 53.2 ± 44.7 | 0.524 |

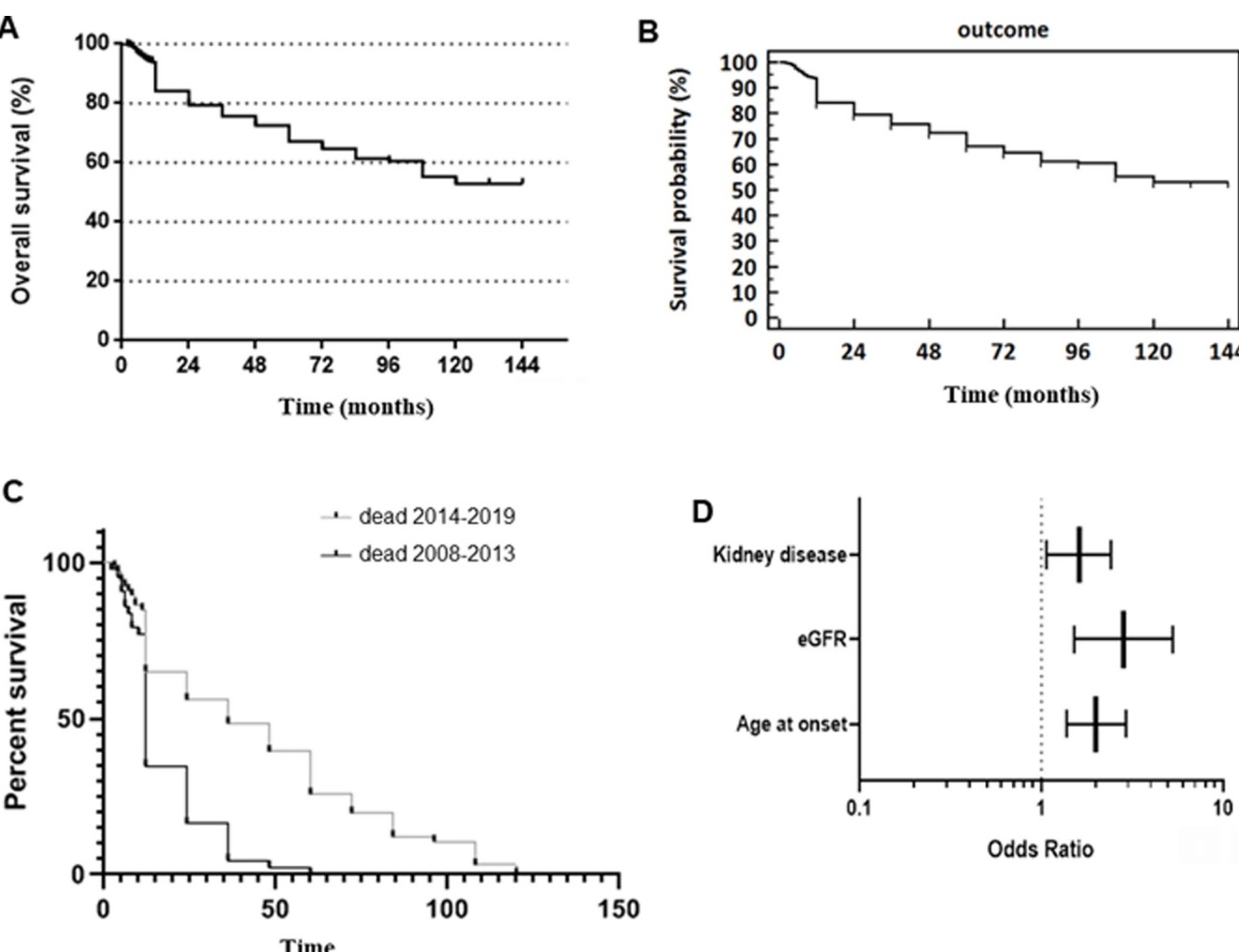

**Fig 1. Kaplan-Meier curves for survival.** A. Overall survival in patients with diabetic foot ulcers. B. Survival probability in patients with diabetic foot ulcers. C. Percent survival in patients with diabetic foot ulcers in the two periods of observation. D. Cox regression multivariate analysis, predictive variables for mortality in patients with diabetic foot ulcers.

diabetic patients CKD remained a significant risk factor for mortality, even after adjusting for other variables. Considering CKD as a surrogate marker for microvascular damage, which in turn is linked to higher risk of neuropathy and vascular disease, both of which are associated with poor wound healing and survival, our results are not surprising but exclude ethnicity playing a role in the combination between renal damage and development of DFU. Indeed, in our study all patients included in the two groups were Sicilian and belonged to a medium or low social class. Moreover, patients with DFU were older and had longer duration of diabetes but these factors, although they do not seem to exert a very important role, remain the principal risk factors. Indeed, adjusted OR (95% CI) detected in diabetic patients developing DFU stressed that older age at disease onset exerts the principal role [13]. Our results suggest that the older the age at which the onset of DFU occurs, the more reduced is the regenerative capacity of the tissues in terms of healing and the higher the possibility of having serious complications leading to death. It is not surprising that increased age is associated with increased mortality, and this has also been shown in other patient populations [3, 29, 32–34]. By contrast, it is difficult to explain the possible role of duration of Type 2 diabetes for developing

DFU, which was not confirmed either in the univariate analysis in the two eras of observation or in multivariate analysis. In this regard it is important to consider that Type 2 diabetic patients who died in both eras were older than those that were still alive.

Many other studies in patients with DFU found that male gender was a risk factor for increased mortality [1, 2, 13, 19]. However, this was not demonstrated by our study, despite a male predominance in our cohort. The reason why males are at increased risk for foot ulceration is still unclear. It has been suggested that men have a higher risk of developing neuropathy as they are taller, and women in the reproductive age group have better endothelial function in their micro- and macro-circulation [1].

This hypothesis has also been taken into consideration for the more frequent renal complications in the male sex of patients with Type 1 diabetes but to this day remains conjectural; perhaps it is linked to different genetic factors in the two sexes, partially demonstrated [35].

Few studies have explored the relationship between DFU and cause-specific mortality [29].

Although a number of risk factors associated with the development of ulceration are well recognized, there is no consensus on which ones dominate, and there are currently no reports of any studies that might justify any specific strategy for population selection in primary prevention [1, 10]. Nevertheless, from our study there is an emerging message, as in the second era we examined DFU patients when an MDFT was created. These data are testified by the fact that multi-dose insulin therapy was maintained during the entire period of hospitalization and the subsequent period required for healing, and above all major attention was dedicated to combined therapies, (i.e. antiplatelet and hypolipidemic therapies, etc), and additionally the grading and stages scores show a tendency towards an amelioration of parameters exerting a role in mortality and overall in amputations [36, 37]. The latter indeed increased in the second era (2014–2019) when the team was created and surgery was more rapid, in particular for minor amputations, in consideration of a complete evaluation of the stages and the Texas grading that was more timely, as demonstrated by the higher distribution of the stages and grades of the DFUs.

This apparent improvement in our study may also be related to the application of the changes following the publication of the National Institute for Health and Care Excellence guidelines on the management of DFUs [7, 38]. Of course, it might be interesting to discover whether the new drugs, as well as GLP-1RA and SGLT-2i, may influence the outcomes of our study, that is to say whether they are able to shorten both the hospitalization and healing periods. In fact, very few data have emerged on the new drugs, strongly suggested in the 2020 guidelines for Type 2 diabetic patients with atherosclerosis cardiovascular disease (ASCD), as regards DFU outcomes [9].

As a secondary outcome we evaluated the differences between the two eras (2008–2013 and 2014–2019). Minor amputations were more frequent in the second era maybe due to the presence in our team of a dedicated surgeon, permitting more rapid intervention in relation to both DFU stages and Texas grading. At the same time major amputations were very few in both periods. Basal bolus insulin was used more in patients hospitalized in 2014–2019, probably because of the new guidelines for DFU treatment and in general because of the new guidelines for management of type 2 diabetes, as confirmed by the fact that in the 2008–2013 period oral hypoglycemic agents were significantly more used.

Dyslipidemia, stroke, cardiac insufficiency and peripheral vascular disease were more frequent in patients who died of the period 2014–2019 compared to 2008–2013, showing that despite the higher frequency of vascular complications, these patients died less than in the second period, supporting that independently from the comorbidities the number of death was lower, maybe due to the improvement in the management of DFU.

## Conclusions

In conclusion, in our study moderate and severe CKD, older age at the onset and reduction of eGFR < 92 ml/min appeared to be the main factors associated with reduced survival of DFU patients. MDFT could be considered promising in the future for more rapid auditing of diabetic foot ulcer numbers, decrease in the number of deaths for diabetic foot ulcers and improvement in the management of this complication.

Patients with DFU have high mortality and reduced life expectancy. Age at diagnosis of diabetic foot ulcer, eGFR values and CKD are risk factors for mortality. The presence of a DFU should be seen by health care providers as an alarming signal of possible premature death, and induce them to initiate intensive risk factor reduction and close follow-up.

## Supporting information

**S1 Table. General characteristics of dead and alive patients with diabetic foot complication divided in the two periods of hospitalization.**
(DOC)

**S2 Table. Clinical, metabolic and inflammatory parameters in all, dead and alive patients with diabetic foot complication divided according to the time of hospitalization.**
(DOCX)

## Author Contributions

**Conceptualization:** Valentina Guarnotta, Stefano Radellini, Enrica Vigneri, Achille Cernigliaro, Felicia Pantò, Salvatore Scondotto, Piero Luigi Almasio, Giovanni Guercio.

**Data curation:** Achille Cernigliaro, Felicia Pantò, Salvatore Scondotto.

**Formal analysis:** Valentina Guarnotta, Stefano Radellini, Enrica Vigneri, Piero Luigi Almasio, Giovanni Guercio.

**Investigation:** Valentina Guarnotta, Stefano Radellini, Enrica Vigneri, Achille Cernigliaro, Felicia Pantò, Salvatore Scondotto, Piero Luigi Almasio, Giovanni Guercio, Carla Giordano.

**Methodology:** Valentina Guarnotta, Stefano Radellini, Enrica Vigneri, Piero Luigi Almasio, Giovanni Guercio.

**Writing – original draft:** Valentina Guarnotta, Stefano Radellini, Enrica Vigneri, Piero Luigi Almasio, Giovanni Guercio.

**Writing – review & editing:** Valentina Guarnotta, Stefano Radellini, Enrica Vigneri, Piero Luigi Almasio, Giovanni Guercio, Carla Giordano.

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
