## [Decision Letter · Decision Letter 0]

21 Jul 2021

PONE-D-21-21103

Diabetic Foot Ulcers: a retrospective analysis of Sicilian patients hospitalized in 2008-2013 and 2014-2019 years

PLOS ONE

Dear Dr. Giordano,

Thank you for submitting your manuscript to PLOS ONE. After careful consideration, we feel that it has merit but does not fully meet PLOS ONE’s publication criteria as it currently stands. Therefore, we invite you to submit a revised version of the manuscript that addresses the points raised during the review process.

We look forward to receiving your revised manuscript.

Kind regards,

Kanhaiya Singh, Ph.D

Academic Editor

PLOS ONE

Additional Editor Comments:

Although the reviewers found this study interesting, they are concerned about some data presentation, interpretation and conclusions drawn in this study.

Journal Requirements:

4. Please include your tables as part of your main manuscript and remove the individual files. Please note that supplementary tables should be uploaded as separate "supporting information" files.

5. We noticed you have some minor occurrence of overlapping text with the following previous publication(s), which needs to be addressed:

- https://bmcendocrdisord.biomedcentral.com/articles/10.1186/s12902-018-0327-2

In your revision ensure you cite all your sources (including your own works), and quote or rephrase any duplicated text outside the methods section. Further consideration is dependent on these concerns being addressed.

Reviewers' comments:

Reviewer's Responses to Questions

**Comments to the Author**

1. Is the manuscript technically sound, and do the data support the conclusions?

Reviewer #1: No

Reviewer #2: Partly

2. Has the statistical analysis been performed appropriately and rigorously? 

Reviewer #1: No

Reviewer #2: I Don't Know

3. Have the authors made all data underlying the findings in their manuscript fully available?

Reviewer #1: Yes

Reviewer #2: Yes

4. Is the manuscript presented in an intelligible fashion and written in standard English?

Reviewer #1: No

Reviewer #2: No

5. Review Comments to the Author

Reviewer #1: In the present article entitled “Diabetic Foot Ulcers: a retrospective analysis of Sicilian patients hospitalized in 2008-

2013 and 2014-2019 years” Giordano and colleagues, have retrospectively analyzed incidence, management and

mortality of DFU in Sicilian Type 2 diabetic patients. Although the manuscript contains extensive amount of work, I have several concerns about data presentation and interpretation. Here are the specific comments.

Title is should be in lines with the overall objective of the study; think title may be changed to 'DFU: improvement in outcomes after implementing standardized care' or 'DFU: retrospective comparative analysis from Sicily between two eras'

Abstract:

Aim: 'lustra', can use standard words like 'era'

Methods: can change to: we compared the two eras, era1: 2008-13, era2: 2014-19. In era 1, n=149, and so on...

Conclusion: advanced age/senility itself is a principal risk factor of death. What was done to remove this bias?

Fig 1 A, B, specify unit on x axis in fig 1, a and b

table 1, typo error: Ischemic, not Hischemic

Methodologies: Ref check: ref 5: International Working Group guidelines on the Diabetic Foot is not same as NIHCE guidelines.: needs reference check.

Reference check again required: ref 26 defines 'Percutaneous transluminal angioplasty and stenting for carotid artery stenosis', the authors may choose to instead choose criteria for Peripheral / femoral angioplasty since we are looking at DFU.

Statistically: it will be more logical to compared dead patients from both eras to arrive at any conclusion regarding effectiveness of changed guidelines other than decreased mortality rates.

Overall, the supplementary data needs to be added for patient details and especially the methodological inconsistencies need to be addressed.

Reviewer #2: Ref: PONE-D-21-21103

In the present article entitled “Diabetic Foot Ulcers: a retrospective analysis of Sicilian patients hospitalized in 2008-

2013 and 2014-2019 years” Guarnotta et al. have retrospectively analyzed incidence of DFU in Sicilian population. Further they have attempted to look at the change in mortality between two time periods. The study design is simple and the results are reasonably by the data. However, some points need to be addressed to make the study robust for the publication.

Major concern: the conclusion does not clearly answer the aim of the study.

line 284: Conclusion: "MDFT could be considered promising in the future for more rapid auditing of diabetic foot ulcer numbers and for establishment of predictive risk factors causing mortality" How do the findings of this study clearly establish the role of a multi-disciplinary team, what exact quality improvement was achieved by such implementation needs to be addressed.

Statistical analysis.to reach any conclusions, the patients who died from era I need to be compared to era II

Minor conclusion:

Authors need to address the "latent period of diabetes before patients reach the clinic"

Title can be improved. “...2008-2013 and 2014-2019 years” needs to be rephrased.

Authors have used “Lustra”. Please stick to use of standard English words.

methods: line 94: International Working Group guidelines on the Diabetic Foot is may not be same as NIHCE guidelines.: needs reference check/ justify

line 187: "who dies higher frequency" can be changed to "that died had". Please fix.

"figure 1: specify unit of time on x axis"

"table 5, typo error: last line: hospitalized in the patients"

overall: use standard abbreviation for number, 'n'

Best regards,

6. PLOS authors have the option to publish the peer review history of their article (what does this mean?). If published, this will include your full peer review and any attached files.

Reviewer #1: No

Reviewer #2: No

---

## [Author Response · Author response to Decision Letter 0]

26 Jul 2021

Reviewer #1: In the present article entitled “Diabetic Foot Ulcers: a retrospective analysis of Sicilian patients hospitalized in 2008-

2013 and 2014-2019 years” Giordano and colleagues, have retrospectively analyzed incidence, management and

mortality of DFU in Sicilian Type 2 diabetic patients. Although the manuscript contains extensive amount of work, I have several concerns about data presentation and interpretation. Here are the specific comments.

Title is should be in lines with the overall objective of the study; think title may be changed to 'DFU: improvement in outcomes after implementing standardized care' or 'DFU: retrospective comparative analysis from Sicily between two eras'

Thanks for this interesting suggestion. We changed the title in “Diabetic Foot Ulcers: retrospective comparative analysis from Sicily between two eras”

Abstract:

Aim: 'lustra', can use standard words like 'era'

Thanks for the comment. We changed lustra with era. 

Methods: can change to: we compared the two eras, era1: 2008-13, era2: 2014-19. In era 1, n=149, and so on...

Thanks for this suggestion. We changed the sentence. 

Conclusion: advanced age/senility itself is a principal risk factor of death. What was done to remove this bias?

Thanks for the comment. We did not remove this bias, because we calculated how much the age at onset of diabetic foot ulcer influence on the risk of death, calculating the odds ratio. 

Fig 1 A, B, specify unit on x axis in fig 1, a and b

Thanks for the comment. We added the units in the figure 1. 

table 1, typo error: Ischemic, not Hischemic

Thanks for the comment. We corrected the typing error.

Methodologies: Ref check: ref 5: International Working Group guidelines on the Diabetic Foot is not same as NIHCE guidelines.: needs reference check.

Reference check again required: ref 26 defines 'Percutaneous transluminal angioplasty and stenting for carotid artery stenosis', the authors may choose to instead choose criteria for Peripheral / femoral angioplasty since we are looking at DFU.

Thanks for the comment. We checked and corrected the references. 

Statistically: it will be more logical to compared dead patients from both eras to arrive at any conclusion regarding effectiveness of changed guidelines other than decreased mortality rates.

Thanks for the interesting suggestion. We compared patients who died in the first era with second era.

Overall, the supplementary data needs to be added for patient details and especially the methodological inconsistencies need to be addressed.

Thanks for the comment. We added supplemental tables. 

Reviewer #2: Ref: PONE-D-21-21103

In the present article entitled “Diabetic Foot Ulcers: a retrospective analysis of Sicilian patients hospitalized in 2008-2013 and 2014-2019 years” Guarnotta et al. have retrospectively analyzed incidence of DFU in Sicilian population. Further they have attempted to look at the change in mortality between two time periods. The study design is simple and the results are reasonably by the data. However, some points need to be addressed to make the study robust for the publication.

Major concern: the conclusion does not clearly answer the aim of the study.

line 284: Conclusion: "MDFT could be considered promising in the future for more rapid auditing of diabetic foot ulcer numbers and for establishment of predictive risk factors causing mortality" How do the findings of this study clearly establish the role of a multi-disciplinary team, what exact quality improvement was achieved by such implementation needs to be addressed.

Thanks for the comment. We added in the conclusion section that multidisciplinary team have had an impact on reducing mortality due to a better management of diabetic foot. 

Statistical analysis.to reach any conclusions, the patients who died from era I need to be compared to era II

Thanks for the comment. We compared patients who died in the two eras (Tables 4 and 5).

Minor conclusion:

Authors need to address the "latent period of diabetes before patients reach the clinic"

Thanks for the comment. We changed it.

Title can be improved. “...2008-2013 and 2014-2019 years” needs to be rephrased.

Thanks we changed the title as suggested.

Authors have used “Lustra”. Please stick to use of standard English words.

Thanks for the suggestion, we changed lustra with eras.

methods: line 94: International Working Group guidelines on the Diabetic Foot is may not be same as NIHCE guidelines.: needs reference check/ justify

Thanks for the suggestion, We corrected the reference.

line 187: "who dies higher frequency" can be changed to "that died had". Please fix.

Thanks for the comment. We modified the sentence.

"figure 1: specify unit of time on x axis"

Thanks for the suggestion. We specified the units in the figure. 

"table 5, typo error: last line: hospitalized in the patients"

Thanks for the comment. We checked and corrected the typing error.

overall: use standard abbreviation for number, 'n'

Thanks for the comment. We used standard abbreviations.

---

## [Decision Letter · Decision Letter 1]

19 Oct 2021

Diabetic Foot Ulcers: retrospective comparative analysis from Sicily between two eras

PONE-D-21-21103R1

Dear Dr. Giordano,

We’re pleased to inform you that your manuscript has been judged scientifically suitable for publication and will be formally accepted for publication once it meets all outstanding technical requirements.

Kind regards,

Kanhaiya Singh, Ph.D

Academic Editor

PLOS ONE

Additional Editor Comments (optional):

Reviewers' comments:

Reviewer's Responses to Questions

**Comments to the Author**

1. If the authors have adequately addressed your comments raised in a previous round of review and you feel that this manuscript is now acceptable for publication, you may indicate that here to bypass the “Comments to the Author” section, enter your conflict of interest statement in the “Confidential to Editor” section, and submit your "Accept" recommendation.

Reviewer #1: All comments have been addressed

Reviewer #2: All comments have been addressed

2. Is the manuscript technically sound, and do the data support the conclusions?

Reviewer #1: (No Response)

Reviewer #2: Yes

3. Has the statistical analysis been performed appropriately and rigorously? 

Reviewer #1: Yes

Reviewer #2: N/A

4. Have the authors made all data underlying the findings in their manuscript fully available?

Reviewer #1: Yes

Reviewer #2: Yes

5. Is the manuscript presented in an intelligible fashion and written in standard English?

Reviewer #1: Yes

Reviewer #2: Yes

6. Review Comments to the Author

Reviewer #1: (No Response)

Reviewer #2: (No Response)

7. PLOS authors have the option to publish the peer review history of their article (what does this mean?). If published, this will include your full peer review and any attached files.

Reviewer #1: No

Reviewer #2: No

---

## [Editor Report · Acceptance letter]

17 Nov 2021

PONE-D-21-21103R1 

Diabetic Foot Ulcers: retrospective comparative analysis from Sicily between two eras 

Dear Dr. Giordano:

I'm pleased to inform you that your manuscript has been deemed suitable for publication in PLOS ONE. Congratulations! Your manuscript is now with our production department. 

Kind regards, 

on behalf of

Dr. Kanhaiya Singh 

Academic Editor

PLOS ONE